# Development of an Automated, Non-Enzymatic Nucleic Acid Amplification Test

**DOI:** 10.3390/mi12101204

**Published:** 2021-09-30

**Authors:** Zackary A. Zimmers, Alexander D. Boyd, Hannah E. Stepp, Nicholas M. Adams, Frederick R. Haselton

**Affiliations:** 1Department of Biomedical Engineering, Vanderbilt University, Nashville, TN 37212, USA; zackary.a.zimmers@vanderbilt.edu (Z.A.Z.); alexander.d.boyd@vanderbilt.edu (A.D.B.); hannah.e.stepp@vanderbilt.edu (H.E.S.); n.adams@vanderbilt.edu (N.M.A.); 2Department of Chemistry, Vanderbilt University, Nashville, TN 37212, USA

**Keywords:** automation, non-enzymatic, DNA amplification, L-DNA, microfluidic, fluorescence

## Abstract

Among nucleic acid diagnostic strategies, non-enzymatic tests are the most promising for application at the point of care in low-resource settings. They remain relatively under-utilized, however, due to inadequate sensitivity. Inspired by a recent demonstration of a highly-sensitive dumbbell DNA amplification strategy, we developed an automated, self-contained assay for detection of target DNA. In this new diagnostic platform, called the automated Pi-powered looping oligonucleotide transporter, magnetic beads capture the target DNA and are then loaded into a microfluidic reaction cassette along with the other reaction solutions. A stepper motor controls the motion of the cassette relative to an external magnetic field, which moves the magnetic beads through the reaction solutions automatically. Real-time fluorescence is used to measure the accumulation of dumbbells on the magnetic bead surface. Left-handed DNA dumbbells produce a distinct signal which reflects the level of non-specific amplification, acting as an internal control. The autoPiLOT assay detected as little as 5 fM target DNA, and was also successfully applied to the detection of *S. mansoni* DNA. The autoPiLOT design is a novel step forward in the development of a sensitive, user-friendly, low-resource, non-enzymatic diagnostic test.

## 1. Introduction

Nucleic acids are among the most important biomarkers of disease, and a large variety of diagnostic nucleic acid tests (NATs) have been developed to detect their presence in diagnostic settings, including polymerase chain reaction (PCR), loop-mediated isothermal amplification (LAMP), and rolling circle amplification [1,2,3,4]. Although very powerful, these tests rely on the use of enzymes, which poses several challenges for use at the point of care or in low-resource settings. For example, enzymes are typically the most expensive reagent in a NAT, and they require robust low-temperature storage conditions and labor-intensive sample preparation methods to achieve high sensitivity [5,6].

In response to these challenges, non-enzymatic NATs have been developed which amplify a target-induced signal using only thermodynamically-driven DNA hybridization reactions. Examples include hybridization chain reaction, entropy-driven catalysis, and catalyzed hairpin assembly [7,8,9,10]. These tests overcome the obstacles posed by enzymes, but typically have sub-optimal sensitivity due to their poor amplification. There have been several interesting demonstrations of non-enzymatic NATs that achieve increased sensitivity via exponential growth, including cascaded catalyzed hairpin assembly [11], branched or hyperbranched hybridization chain reaction [12,13], and dendritic amplification [14], but perhaps the most promising is the recent demonstration of a dumbbell DNA amplification scheme [15]. Each DNA dumbbell, shown in Figure 1, has four binding domains which bind the opposite dumbbell. The original authors performed the amplification assay by capturing target DNA on magnetic beads and sequentially incubating the beads with U1, then U2, then U1, etc. Over the course of 35 dumbbell incubations, a limit of detection of 5 copies/reaction was reported; this far surpasses limits of detection typical of other non-enzymatic NATs.

A serious drawback of this work which was not discussed is the amount of time, labor, and reagents required. Due to the single-stranded complementary binding domains, combining the two dumbbells in solution will cause uncontrolled binding to one another, rather than controlled accumulation on the surface of the magnetic beads. Therefore, the dumbbells must be added to the beads one at a time, with wash steps in between each incubation. To perform 35 successive dumbbell incubations, as well as the initial target capture step, approximately 18 h were required. Over the course of those 18 h, several pipetting steps were required every 30 min, each one a new opportunity for human error. These challenges are exacerbated by the lack of an internal negative control; a second parallel reaction was required to measure non-specific amplification, doubling the total amount of labor, time, and reagents required. These two obstacles, the undesirably high number of hands-on steps and the lack of an internal negative control, make the dumbbell DNA assay impractical for diagnostic applications. If the difficulties of performing this assay can be alleviated, it would have all the qualities of a powerful NAT appropriate for use in low-resource settings.

Previous works have pioneered the use of pre-arrayed reaction cassettes as a simplified and effective method of magnetic sample processing [16,17,18]. In this approach, different solutions are loaded into microfluidic tubing and separated by air gaps. The surface tension of these solutions maintains the integrity of the air gaps, and sealing the ends of the tubing immobilizes the contents. Magnetic beads are then transported through the different solutions via movement of an external magnet. Inspired by these methods, we designed an automated reaction processor to perform the dumbbell amplification assay without the need for repeated human pipetting steps. After the initial target capture step, the magnetic beads are loaded into a reaction cassette along with each dumbbell, and automatic movement of the tubing relative to a stationary magnetic field performs the steps previously performed manually. Fluorescent labels on the dumbbells create a real-time optical readout that can be measured throughout the reaction.

To eliminate the need for a second control reaction, left-handed dumbbells with their own distinct fluorescent labels are included as a built-in negative control. Left-handed DNA (L-DNA) is the chiral enantiomer of right-handed DNA (D-DNA). Although only D-DNA is found in nature, advancements in chemical synthesis techniques have enabled the commercial production of synthetic L-DNA. L-DNA is identical in chemical composition, and exhibits identical solubility, thermodynamic properties, and binding behavior as its right-handed counterpart [19,20,21]. Having previously demonstrated the utility of L-DNA as a built-in control for non-enzymatic DNA circuits [10], we used them here as a measure of non-specific amplification in the dumbbell amplification assay.

Finally, as a proof of principle of the potential diagnostic applications for this automated, non-enzymatic DNA amplification reaction, we demonstrated the detection of *Schistosoma mansoni* DNA. *S. mansoni* is the most widespread member of the family of parasitic flatworms which cause schistosomiasis, a leading neglected tropical disease responsible for the loss of 4.5 million disability-adjusted life years (DALYs) [22,23,24]. The target sequence is part of a 121 bp tandem repeat sequence which comprises approximately 12% of the *S. mansoni* genome, and has previously been targeted with PCR assays [25,26,27]. The PCR limit of detection was reported to be 1.28 pg/mL [27]; given the highly-repeated nature of the target sequence in the genome, this corresponds to approximately 790,000 copies/mL of the target sequence.

## 2. Materials and Methods

### 2.1. Oligonucleotides

All D-DNA sequences were purchased from Integrated DNA Technologies (Skokie, IL, USA), and all L-DNA sequences were purchased from Biomers (Ulm, Germany). Each dumbbell component was modified with a 5′ FAM for D-DNA or a 5′ Texas Red (TXR) for L-DNA. All fluorescently-labeled oligonucleotides were HPLC purified. A complete list of DNA sequences is given in Table 1, with distinct binding domains separated by underscores. Oligonucleotides were suspended in tris/EDTA (TE) buffer at a concentration of 100 µM and stored at −20 °C for long-term storage. For short-term storage, subsequent aliquots were created by diluting to 4 µM in 2X saline sodium citrate (SSC) buffer, and stored at 4 °C.

The dumbbell sequences (first 7 in Table 1) were adapted from a previous work by Xu et al. [15]. Sequences original to this work (last 5 in Table 1) are those designed to detect *S. mansoni* DNA, as well as ‘U1-a* tag’ and ‘U1* removal’, which were designed for a toehold-mediated strand displacement system to separate the dumbbells from the magnetic beads.

### 2.2. Design of the AutoPiLOT Reaction Processor

The automated Pi-powered looping oligonucleotide transporter (autoPiLOT) is an automated reaction processing device controlled by a Raspberry Pi B 3+ microcomputer. The Pi is connected via USB to two Arduino Unos, called the sensor Arduino and motor Arduino. Both Arduinos are connected to a 6 mm tactile button that, when pressed by the user, activates different segments of their onboard code. A circuit diagram is shown in Appendix A. The master code for the Raspberry Pi, as well as the onboard code for both Arduinos, are available upon request.

Two infrared reflective object sensors (Digi-Key, QRD1114) are connected to the sensor Arduino for detection of liquid/air interfaces. A stepper motor (Applied Motion Products, STR4 drive, HT23 motor) is connected to the motor Arduino, which turns gears controlling the movement of the microfluidic tubing. Fluorinated ethylene propylene (FEP) tubing—also commonly called PTFE tubing—with 1/8” outer diameter and 3/32” inner diameter (McMaster-Carr, 9369T24) was used to house the reaction contents. 5/8” × 5/8” × 1/4” neodymium magnets (K&J Magnetics Inc, BAA4) were used to control the magnetic beads. A Qiagen ESElog fluorometer was used for fluorescence measurements. Housing for the tubing, magnets, and fluorometer was printed using a MarkForged Mark 2 3D printer, and the 3D drawing is also available upon request.

The autoPiLOT is shown in Figure 2. The housing was 3D printed to hold the magnets, IR sensors, fluorometer, and microfluidic tubing. Not pictured is the Raspberry Pi microcomputer and Arduinos which control the device. The low cost and portability of these components make this device easily transportable and independent of a laptop or other computer for operation. The microfluidic tubing is held between the gears, which are controlled by a stepper motor. Rotation of the gears therefore moves the microfluidic tubing which contains the various reaction fluids, as previously described for one-directional sample prep devices [28,29]. The reaction cassette was moved at a speed of 0.2 cm/s to transport the magnetic beads, and 7.5 cm/s to break the beads out of the magnetic field.

To account for variations in air gap size or fluid volume during the preparation of the reaction cassette, a pair of infrared sensor/receivers was interfaced with the system to detect the locations of the liquid/air interfaces in the tubing. Before performing the cyclical amplification reaction, a “reconnaissance run” is performed which detects changes in IR transmittance as the tubing moves from one end to the other. These changes in transmission indicate a change from liquid to air, and the locations and sizes of each fluid chamber are mapped using this information. Because it uses these measurements to guide the movement of the tubing, the autoPiLOT reaction is adaptable to variations in the pre-loaded reaction cassettes.

The movement of the magnetic beads between chambers is a result of the stationary magnetic field holding the beads as the tubing is moved by the gears. Tween 20 decreases the surface tension of the fluid chambers, which allows the beads to break through the interface. Several different concentrations of Tween were tested to see if they impacted the binding of the dumbbells to one another, and the results (shown in Appendix A) show that none inhibited binding. Since dumbbell binding appeared equally efficient for all tested buffer conditions, 0.05% Tween in 5X SSC buffer was used in accordance with the protocol for the original dumbbell assay [15].

The movement of the magnetic beads through the four reaction chambers (shown in Figure 3) is as follows: from their starting place in the beads chamber, they are carried into U2. Here, the beads are mixed and then removed from the magnetic field (via a fast movement speed) and then incubated for 30 min. The beads are then recollected in the magnetic field and carried into the wash chamber, where they are dispersed across the length of the chamber. The wash chamber is moved across the optical path of the fluorometer for fluorescence readings, and then the beads are transported into U1. Again, they are mixed and incubated to bind dumbbells. The beads are then moved in the reverse direction to U2, and the cycle begins again. Fluorescence measurements are collected in the wash chamber after dumbbell incubations number 1, 3, 5, etc. All reaction steps after the incubation with dumbbell U1* are therefore automated.

### 2.3. Gel Electrophoresis Studies

A series of experiments were conducted using agarose gel electrophoresis to investigate whether the predicted DNA hybridization events were taking place. Gels contained 3% ultra-pure agarose (Thermo Fisher) in 0.5X tris/borate/EDTA (TBE) buffer, and were stained with 1X GelRed nucleic acid stain (GoldBio, originally 10,000X in water). An ultra-low range DNA ladder (ThermoFisher) was included in each gel at a concentration of 2 ng/µL. Gels were run at 60V for approximately 1 h, then imaged using a Bio-Rad Gel Doc EZ Imager.

### 2.4. Dumbbell Formation

To form the double-stranded dumbbell structures, the two oligonucleotides (for example, U1-a and U1-b form dumbbell U1) were combined at equal molar ratios to the desired final concentration and then heated with the following thermal profile: 95 °C for 5 min, followed by 50 °C for 10 min, and finally by 37 °C for 10 min. This process was performed in a Qiagen Rotor-Gene PCR thermal cycler. The dumbbells were then stored at room temperature until use. For experiments using both D-DNA and L-DNA dumbbells, both D-DNA and L-DNA were combined in the same tube. For example, D-DNA U1-a and U1-b were combined with L-DNA U1-a and U1-b in the same tube for a final solution of right- and left-handed U1.

### 2.5. Magnetic Bead Functionalization

Dynabeads MyOne Streptavidin T1 magnetic beads (ThermoFisher) were functionalized with a biotinylated capture probe designed to bind the DNA target of interest. First, the beads were washed three times using hybridization buffer (5X SSC buffer + 0.05% Tween 20). Each wash consists of magnetically separating the beads from solution and, while separated, removing the supernatant and adding fresh buffer. After washing, the beads were resuspended in the biotinylated capture probe at a ratio of 2.5 nanomoles DNA per mg beads, briefly vortexed, and incubated on a rotisserie for 20 min. The beads were then washed three times with hybridization buffer and resuspended in blocking buffer (1X PBS + 16 µM free biotin + 0.05% Tween 20) and incubated on the rotisserie for 30 min. Experiments investigating other blocking methods used blocking buffers in which the free biotin was replaced with 2% bovine serum albumin (BSA) by volume, 200 µg/mL salmon sperm DNA, or removed altogether. The beads were then washed three times with hybridization buffer and stored in stocking buffer (1X PBS + 0.05% Tween 20) at a final concentration of 1 mg/mL. D-Biotin and BSA were purchased from Thermo Fisher, and Tween 20 was purchased from Sigma Aldrich.

### 2.6. Dumbbell Amplification Reactions

An amount of 50 µg of functionalized magnetic beads was used in each reaction. The beads were washed three times in hybridization buffer and resuspended in 100 µL of sample solution. The solution was vortexed to disperse the beads throughout and incubated on a rotating rotisserie. Sequence-specific nucleic acid capture via functionalized magnetic beads has previously been shown to be a slow process, with low efficiency for incubation times shorter than 30 min [17]. Therefore, 30 min was chosen as the incubation time. Next, 10 µL of U1* at 1 µM was added, and the beads incubated for another 30 min on the rotisserie. The beads were then washed three times in hybridization buffer, and resuspended in 75 µL hybridization buffer. At this point, the functionalized beads have bound target, which has in turn bound the first dumbbell U1*. Incubations in dumbbells U1 and U2 were then performed automatically using the autoPiLOT reaction processor.

The microfluidic FEP tubing was loaded with the following fluid chambers, each separated by a small air gap approximately 1 cm in length: 100 µL U1 at 250 nM, 100 µL hybridization buffer, 100 µL U2 at 250 nM, 100 µL magnetic bead solution. To load a fluid chamber into the tubing, the liquid was pipetted directly into the tube with the pipette tip flush against the opening. The tubing was then gently tipped to move the fluid chamber further along, clearing room for the next solution to be inserted. After the four fluid chambers were loaded in the tubing, both ends were sealed with Cha-seal tube sealing compound. The loaded tube (shown in Figure 3) was run on the autoPiLOT to move the magnetic beads back and forth between the two dumbbell solutions, reading fluorescence on the beads in the wash chamber after every other dumbbell incubation. The total reaction time was a function of the number of dumbbell incubations; 15 incubations on the autoPiLOT took approximately 8 h (15 incubations × 0.5 h/incubation, plus the time to move the beads back and forth between chambers).

## 3. Results

### 3.1. Validation of DNA Hybridization Events

Each dumbbell consists of two partially-complementary oligonucleotides; Figure 1 shows a full breakdown of each dumbbell and its components. Briefly, dumbbells U1 and U2 consist of U1-a, U1-b, U2-a, and U2-b, where a and b are used to denote the two strands in each dumbbell. U1* is a modified version of U1 with an extra binding domain on the 5′ end of U1-a to bind to the target DNA. U1* serves as the bridge between the target DNA and the network of dumbbells that forms during the amplification assay. Gel electrophoresis was used to validate that the individual dumbbell strands bind to form dumbbells as predicted.

For each of the three dumbbells, the respective single-stranded components were combined at equal molar ratios and visualized on a gel to confirm hybridization into double-stranded dumbbells; these results are shown in Figure 4A. With the exception of U1-a*, which is 60 bases, all of the single-stranded components are 45 bases in length. These can be seen as the lower bands on the gel. For dumbbells U1 and U1* (Figure 4A, left lanes 3 and 6), the lower band disappears when the two single-stranded components are combined and a new band is formed corresponding to 90 bases, showing that the two strands have hybridized to form the double-stranded dumbbell. The formation of dumbbell U2 appears incomplete, as there is still a prominent band at 45 bases indicative of remaining single-stranded DNA (Figure 4A, right lane 3). Even when different ratios of U2-a and U2-b were used, this extra band persisted (Appendix A). The prominent band at 90 bases, however, is evidence that the dumbbell U2 is still forming, even if not quite as efficiently as U1 and U1*.

Next, the binding of the dumbbells U1 and U2 to one another was examined. Figure 4B shows that once combined at equal ratios, the dumbbells form products of varying sizes. Distinct bands corresponding to complexes of 2, 3, and 4 dumbbells can be seen; the bands resulting from even larger complexes blend together to form a smear on the gel. This test was duplicated using FAM-labeled DNA to examine whether the presence of fluorescent labels had any apparent effect on dumbbell binding. The results (Figure 4B, right) show that the FAM-labeled dumbbells hybridized in a similar fashion as the unlabeled dumbbells. These findings validate three key assumptions moving forward: (1) the dumbbells form as expected from their single-stranded components, (2) the dumbbells bind to each other as expected to form large dumbbell complexes, and (3) attaching fluorescent labels to these dumbbells does not inhibit their affinity for one another.

After validating the D-DNA dumbbell binding events, the previous experiments were repeated using left-handed DNA. The L-DNA sequences are completely identical, the only differences are the chirality of the nucleotide bases and the replacement of FAM with Texas Red. As expected, the results (Appendix A) show that the L-DNA dumbbells exhibit the same behavior as the D-DNA; the dumbbells assemble from their single-stranded components and bind to each other to form large dumbbell networks.

### 3.2. Validation of Target Capture by Magnetic Beads

There are two other key DNA binding steps in the dumbbell DNA amplification assay which were not examined in the previous gel experiments: the binding of target DNA to the capture probe-functionalized beads, and the binding of U1* to the target. Both of these binding events were also examined via gel electrophoresis. The results, shown in Figure 4C, show that the capture and target strands (lanes 1 and 2) hybridize to form a capture-target complex (lane 3). The addition of U1* (lane 4) then creates a capture-target-U1* complex (lane 5). This suggests that the magnetic beads which have been modified with capture probe will capture the target DNA out of solution, and subsequently bind the modified dumbbell U1*. The presence of U1* provides a starting point for cyclical dumbbell amplification. Microscopy was also used to confirm that the fluorescently-labeled U1* was indeed attached to the magnetic beads. Appendix A show beads which have been incubated in target DNA, followed by the FAM-labeled U1*, using brightfield imaging and fluorescence imaging. The fluorescence image shows that FAM (in green) has attached to the beads, which confirms that the binding steps shown in Figure 4C also occur when bound to the surface of magnetic beads.

### 3.3. Fluorescence Measurements Reflect the Amount of DNA on the Beads

One of the features that makes quantitative PCR a gold standard NAT is the quantitative information gained by real-time fluorescence measurements. Fluorescence is read during each thermal cycle to create fluorescence vs. cycle data, and these data can be used to estimate the starting amount of target DNA. To achieve real-time fluorescence measurements in the autoPiLOT assay, fluorescence was measured directly on the surface of the beads. Capture probe-functionalized magnetic beads were incubated with varying amounts of FAM-labeled target DNA, shown in black in Figure 5A. The resulting fluorescence curve, shown in Figure 5B, demonstrates that the autoPiLOT fluorescence readings are directly proportional to the amount of DNA attached to the beads. Next, the same experiment was performed using unlabeled target DNA and followed by incubating with FAM-labeled U1*, as shown in Figure 5B. Just as each target had one FAM fluorophore in the previous design, each target should bind one FAM-labeled U1* in this design. The results (Figure 5B, red points) show that the two fluorescence curves are approximately equal at lower concentrations of target DNA. The U1* fluorescence tapers off at higher concentrations compared to that of the target. One possible explanation for this discrepancy is that when the beads already have a large amount of their surfaces covered in target DNA, the binding of additional DNA is inhibited by steric hindrance [30]. If true, this would suggest that the amplification observed in the autoPiLOT assay will also taper off as the bead surfaces become too overcrowded.

### 3.4. Performance of Parallel L-DNA Dumbbells

The amplification behavior of L-DNA control dumbbells was measured to validate that it was an accurate measure of the non-specific amplification of the D-DNA dumbbells. The fluorometer in the autoPiLOT reaction processor has two fluorescent channels, one that detects FAM on the D-DNA and one that detects Texas Red on the L-DNA. The first step was to ensure that the intensities of these two channels were adjusted such that D-DNA dumbbells produced a signal equal to that of L-DNA dumbbells. The gain of the Texas Red channel was adjusted until the fluorescent signal was equal across multiple dumbbell concentrations (see Figure 6A). Moving forward, the magnitudes of fluorescent measurements are assumed to be directly comparable between D-DNA and L-DNA; matching measurements indicate matching amounts of dumbbells.

Next, negative control reactions were performed using the autoPiLOT. No target was bound by the beads, so only non-specific amplification was observed. The fluorescent measurements for both D-DNA and L-DNA dumbbells over the course of 15 dumbbell incubations are shown in Figure 6B. The amplification of both enantiomers is virtually identical; analysis using two-way analysis of variance (ANOVA) found that the chirality of the dumbbells had no significant effect on the observed variance (α < 0.05). These findings confirm the hypothesis that the amplification of the left-handed dumbbells matches the non-specific amplification of the right-handed dumbbells.

Finally, autoPiLOT amplification was compared between samples with and without L-DNA dumbbells included. Samples contained 3*10^11^ copies of target to ensure that amplification was high, and any differences in performance caused by the presence of the L-DNA would be exaggerated. Over the course of 15 dumbbell incubations, the D-DNA amplification (shown in Figure 7) was found to be identical according to two-way ANOVA, α < 0.05, regardless of whether or not L-DNA dumbbells have been included. The parallel L-DNA dumbbells exhibit two key traits: they amplify at a rate equal to the non-specific amplification of the D-DNA dumbbells, and they do not impact the amplification of the D-DNA dumbbells. These findings demonstrate the utility of a dual-chirality design as a means to measure both target-induced amplification and non-specific amplification in a single reaction.

### 3.5. AutoPiLOT Performance and Limit of Detection Studies

To determine the analytical limit of detection for the autoPiLOT reaction, serial dilutions of target DNA ranging from 3 × 10^11^ to 3 × 10^5^ copies per reaction were amplified for 15 cycles of 30 min incubations. The D-DNA amplification curves for several of these target concentrations are shown in Figure 8. Analysis using two-way ANOVA revealed that both target copy number and number of dumbbell incubations had significant effects on the resulting fluorescence measurements. The slopes of the amplification curves increase with increasing target DNA, resulting in increased fluorescent intensity, especially at later cycles.

To interpret the results of a dual-chirality dumbbell amplification reaction, the signals from both the D-DNA and L-DNA dumbbells were compared. This eliminates the need to perform no-target control reactions. The change in fluorescent signal can be calculated by simply subtracting the first fluorescent measurement from each subsequent measurement; the ratio of change in D-DNA to change in L-DNA was used to compare the two signals. Let this ratio be called the signal ratio. The signal ratio changes with each new fluorescent measurement, but in general tends to increase with additional incubations. Figure 8 shows the signal ratio after 15 dumbbell incubations as a function of target copy number.

Diagnostically, to determine whether a given sample is positive, there must be a threshold signal ratio; an observed signal ratio greater than this threshold is interpreted as a positive result. A ratio of 1 may seem like a reasonable threshold, since D-DNA and L-DNA amplification are expected to be equal in the absence of target DNA. However, there is a certain level of expected noise that means a negative sample may still exceed a signal ratio of 1. To determine a threshold, three autoPiLOT reactions were performed on separate days, and the formula $\mu_{0}+3\sigma_{0}$ was used, where $\mu_{0}$ and $\sigma_{0}$ represent the mean and standard deviation of the signal ratio. Using this calculation, a signal ratio of 1.30 was determined as the threshold to determine positive results. This value is plotted as a red dashed line in Figure 8, and intersects with the linear line of best fit at 3 × 10^5^ copies. Thus, the limit of detection of this reaction was determined to be approximately 3 × 10^5^ copies/reaction, or 5 fM (pink line in Figure 8).

### 3.6. Application of the AutoPiLOT to Detection of S. mansoni DNA

The previous experiments have all used the originally-published development target sequence (see Table 1, target) to test the autoPiLOT platform on well-established DNA sequences. To apply this platform in a diagnostic setting, several DNA elements must be redesigned to integrate a new target sequence. As an example, DNA from the genome of *Schistosoma mansoni* was chosen as a new target. The stepwise binding motif of the dumbbell DNA assay means that dumbbells U1 and U2 do not need to be redesigned to integrate a new target sequence; only the target-binding domain on U1-a* and the capture sequence need to be changed to bind the desired target. New magnetic beads were functionalized using the *S. mansoni* capture probe, and the autoPiLOT assay was performed on samples containing 3 × 10^7^ copies of the *S. mansoni* target. The results are shown in Appendix A alongside the results for the original target sequence. The signal ratios after 15 cycles were calculated as 2.21 ± 0.09 for the original target and 2.18 ± 0.23 for the *S. mansoni* target, which are not significantly different based on a two-way *t*-test, α < 0.05. These results suggest that the autoPiLOT assay can be applied to new targets without sacrificing performance. Although the limit of detection was not determined for the new target, the identical performance at a concentration of 3 × 10^7^ copies/reaction suggests that it would be comparable to previously-determined limit of 3 × 10^5^ copies/reaction.

## 4. Discussion

Real-time fluorescence measurements were shown to reflect the amount of DNA which had accumulated on the magnetic beads. Signal amplification was approximately linear, and the slope increased with increasing amounts of target DNA in the sample, as shown in Figure 7. The autoPiLOT assay exhibited a limit of detection of 5 fM, and showed promising results when applied to the detection of *S. mansoni* DNA, demonstrating the potential for a sensitive, low-cost, point-of-care non-enzymatic NAT.

The observed linear amplification is not in agreement with the previously-reported exponential dumbbell DNA amplification. This discrepancy is likely due to a combination of factors. When the beads are transported into a new chamber in the autoPiLOT, they are briefly mixed back and forth to disperse them across the length of the chamber. Despite this, the beads quickly settle to the bottom of the tubing where they remain for most of the 30 min. Dumbbell binding efficiency would likely improve if the beads were homogenously dispersed throughout the chamber; the effective exposed surface area of the beads would be much larger. Another potential cause of decreased amplification is steric hinderance from the surrounding dumbbells. The more dumbbells are attached to the bead surface, the more chaotic and intertwined the growing network of dumbbells becomes. This would result in amplification which becomes less efficient the more dumbbells are bound; this behavior is suggested in the amplification curves shown in Figure 8. At lower fluorescence values (in the 100–250 A.U. range), the amplification has some concavity indicative of exponential amplification. This is most visible for the 3 × 10^5^ and 3 × 10^7^ copy number curves. As the reaction progresses and more dumbbells attach to the beads, the curve becomes linear, and even begins to taper off at high enough values. Future work should focus on optimization of the autoPiLOT magnetic bead control strategies, in an effort to more efficiently bind dumbbells and achieve exponential amplification.

Future research should also work toward the implementation of a lower-cost fluorescence imaging system. The fluorometer used in the autoPiLOT is the only expensive component, with a price tag of several thousand dollars. Recently, smartphone-based fluorescence microscopes have grown in popularity and sophistication, bringing high quality digital cameras and multifunctionality to the point of care [31,32]. It is estimated that approximately 80% of the world population uses smartphones, making a smartphone optical readout highly applicable, even in low-resource settings [33].

An alternative strategy to avoid use of a costly fluorometer would be a colorimetric readout. The original demonstration of the dumbbell amplification assay utilized a colorimetric readout, in which dumbbells were labeled with biotin, and avidin-labeled horseradish peroxidase (HRP) was added to the beads at the end of the end of the reaction [15]. The drawback of this strategy is that it prohibits the use of two distinct readouts, one for D-DNA and one for L-DNA. The use of a dual-chirality design allowed for the monitoring of not only target signal, but also non-specific amplification in each reaction. Switching to a colorimetric readout simplifies the autoPiLOT components, but sacrifices the innovative dual-chirality functionality.

Reaction automation in the autoPiLOT removes the need for many hands-on pipetting steps, but does not by default decrease the total reaction time. Future research may also investigate shortening the overall autoPiLOT reaction time. We have shown (in Appendix A) that the incubation period in each dumbbell can be greatly reduced while still observing signal amplification. The exact effect on signal-to-noise ratio, however, remains unknown. Further testing of several incubation times to determine a balance between reaction time and sensitivity would be useful for diagnostic applications in which time is limited. Additionally, a more flexible design in which the reaction runs only as long as necessary could be developed; samples with higher target concentrations would amplify quicker, terminating the reaction quicker than samples with less target.

Finally, the performance of the autoPiLOT diagnostic platform should be evaluated using real urine samples. The preliminary data presented here suggest that the autoPiLOT is an effective strategy to detect *S. mansoni* DNA, which has previously been detected in urine samples via PCR. The autoPiLOT demonstrated a limit of detection of 3,000,000/mL target copies, nearly as sensitive as previously described PCR assays. It is possible that reaction sensitivity will decrease when applied to DNA extracted from urine, since urine is known to contain inhibitors such as nucleases. In the case of nucleases, heating the urine to 75 °C for 10 min has been shown to deactivate nucleases in urine [10]. Sample-prep steps such as heating are likely required to maximize extraction efficiency from urine. Losses due to extraction efficiency may be countered by increasing sample volume; large volumes are easily obtained. These characteristics suggest that the autoPiLOT has the potential to be a highly-sensitive, low-resource, non-enzymatic NAT, and the first of its kind.

## 5. Conclusions

The dumbbell DNA amplification scheme has previously been shown to be a highly-sensitive, non-enzymatic method of detecting target DNA. The largest obstacles to its use in diagnostic applications were the extremely high number of manual pipetting steps and hands-on time required, and the need for an additional control reaction. The autoPiLOT platform has overcome these obstacles through automation of the reaction in a self-contained, microfluidic reaction cassette. The components are primarily 3D-printed parts and low-cost electronics, making the autoPiLOT a low-resource compatible diagnostic platform.

The autoPiLOT assay also demonstrates the utility of left-handed L-DNA as an internal control. Whereas a traditional assay format would require the performance of two parallel reactions, one with and one without target, for interpretation of results, the dual-chirality design used here discretely measured both specific and non-specific signals in the same reaction. In addition to saving time and reagents, this opens the door to testing unpurified biological samples which may exhibit varying rates of non-specific amplification that cannot be accurately simulated with a parallel control reaction.

## Figures and Tables

**Figure 1 micromachines-12-01204-f001:**
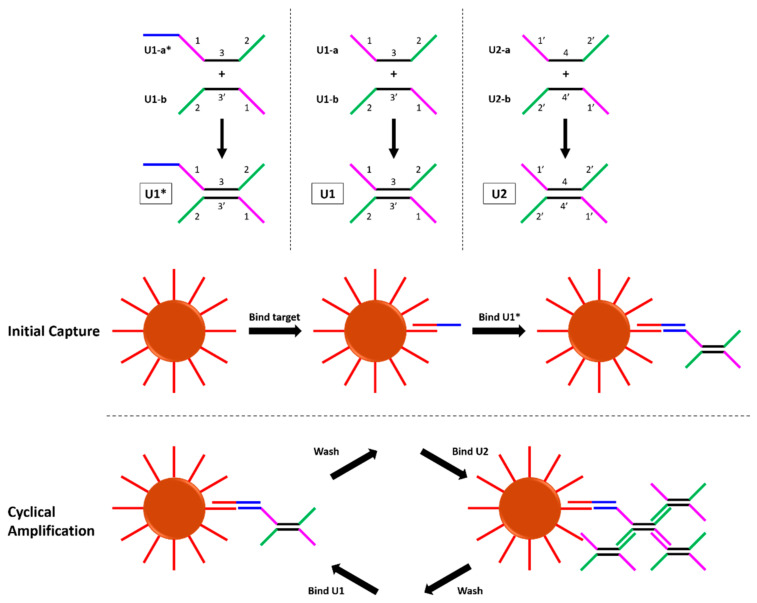
Overview of the dumbbell amplification assay. (**Top**) The DNA dumbbells used in this study. Complementary binding domains are denoted as 1, 1′. (**Bottom**) Binding steps in the dumbbell amplification assay. After binding target and U1*, the beads cycle between incubations of U1 and U2, with intermediate wash steps.

**Figure 2 micromachines-12-01204-f002:**
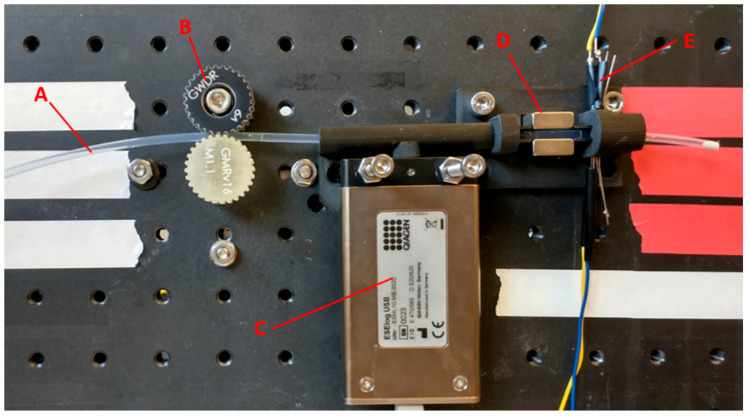
Photograph of the autoPiLOT reaction processor. Microfluidic polymer tubing (A) contains the reaction solutions. Two gears (B) controlled by a stepper motor move the tubing. A fluorometer (C) measures the fluorescence on the beads inside a light-tight chamber. Two neodynium magnets (D) hold the magnetic beads in their field. An IR sensor/receiver pair (E) detects the liquid/air interfaces in the tubing.

**Figure 3 micromachines-12-01204-f003:**
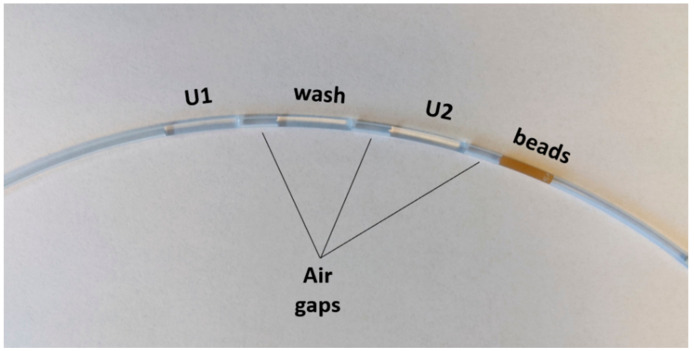
Picture of a pre-arrayed reaction cassette for use in the autoPiLOT reaction processor.

**Figure 4 micromachines-12-01204-f004:**
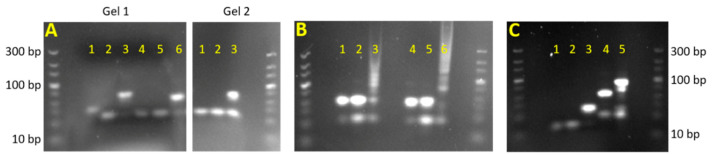
Gel electrophoresis results for D-DNA dumbbell binding studies. The same DNA ladder is used throughout, and shown at the right and left sides of each gel. (**A**) Gel 1, lanes 1-3: U1-a*, U1-b, and U1*. Gel 1, lanes 4-6: U1-a, U1-b, and U1. Gel 2, lanes 1-3: U2-a, U2-b, and U2. (**B**) Lanes 1–3: U1, U2, and U1+U2. Lanes 4–6: identical, but with 5′ FAM fluorescent labels. (**C**) Lane 1: capture probe. Lane 2: target. Lane 3: capture+target. Lane 4: U1*. Lane 5: capture + target + U1*.

**Figure 5 micromachines-12-01204-f005:**
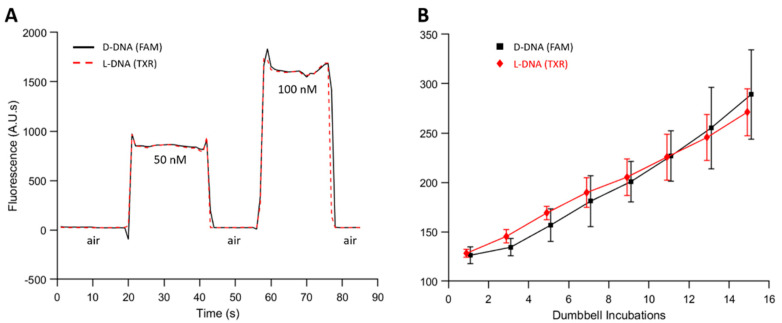
Matching D-DNA and L-DNA fluorescent signals in the autoPiLOT. (**A**) Reaction chambers containing D-DNA and L-DNA dumbbells at 50 and 100 nM were passed through the optical path of the autoPiLOT fluorometer. (**B**) No-target control reactions performed on the autoPiLOT measuring both D-DNA (black) and L-DNA (red) fluorescence. The mean of three trials ± one standard deviation is shown.

**Figure 6 micromachines-12-01204-f006:**
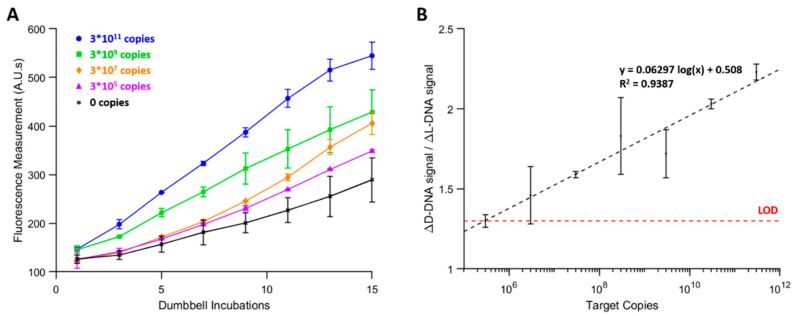
(**A**) D-DNA amplification curves for autoPiLOT trials for a range of target copy numbers. (**B**) Signal ratio, or the ratio of change in D-DNA signal to change in L-DNA signal after 15 dumbbell incubations, is plotted (black dots) as a function of target copy number. A logarithmic line best fit is overlaid as a black dashed line. The red dashed line represents the threshold signal ratio used to determine the limit of detection.

**Figure 7 micromachines-12-01204-f007:**
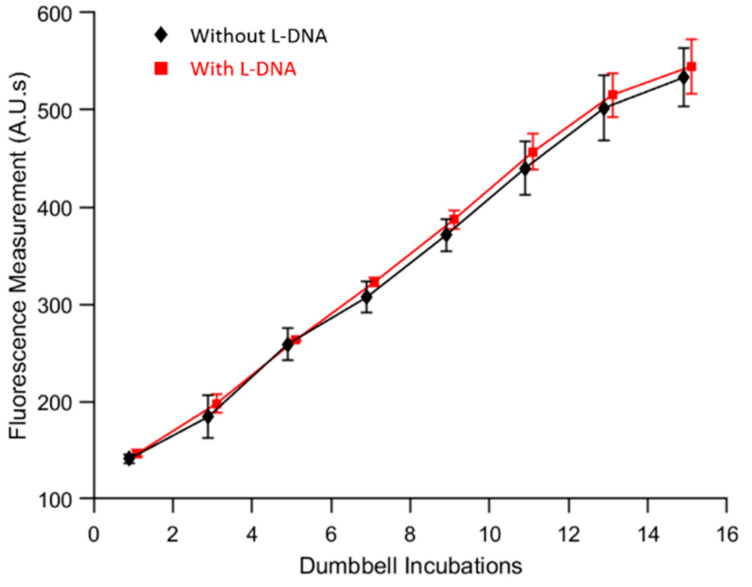
Comparison of D-DNA amplification with (red) and without (black) parallel L-DNA dumbbells included. The mean of three trials ± one standard deviation is shown.

**Figure 8 micromachines-12-01204-f008:**
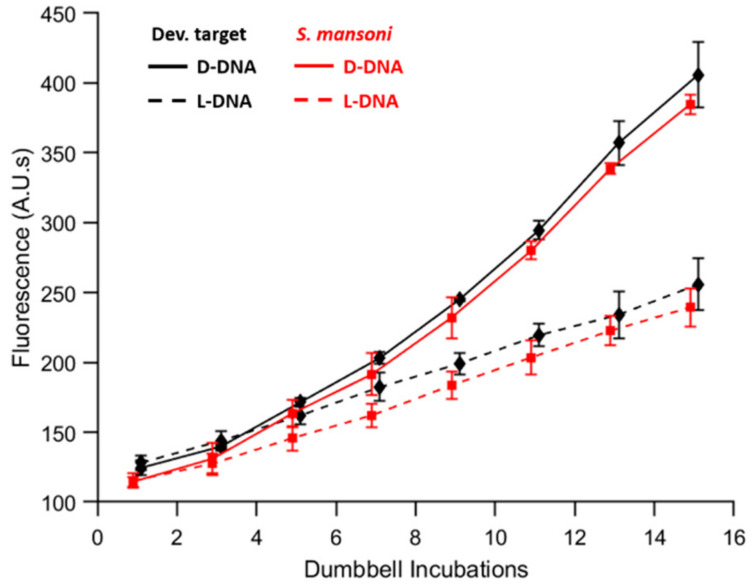
Comparison of autoPiLOT performance for detection of the development target sequence (black) and the *S. mansoni* target sequence (red). D-DNA signals are shown as solid lines, and L-DNA signals as dashed lines. The mean of three trials ± one standard deviation is shown.

**Table 1 micromachines-12-01204-t001:** Oligonucleotide sequences used in this work.

Name	5′ Mod	Sequence (5′–3′)
U1-a	FAM/TXR	CTAGCTCATACATC_ATCCTATCTATCCAGAC_ TCTCACACGTACTC
U1-a*		TCGCTCTTACAAGGCA_CTAGCTCATACATC_ ATCCTATCTATCCAGAC_TCTCACACGTACTC
U1-b	FAM/TXR	CTAGCTCATACATC_GTCTGGATAGATAGGAT_ TCTCACACGTACTC
U2-a	FAM/TXR	GATGTATGAGCTAG_GAGATGCAATCGACTGT_ GAGTACGTGTGAGA
U2-b	FAM/TXR	GATGTATGAGCTAG_ACAGTCGATTGCATCTC_ GAGTACGTGTGAGA
Capture	Biotin	TTTTTTTTTT_CTCATTCACCTACG
Development target		TGCCTTGTAAGAGCGA_CGTAGGTGAATGAG
U1-a* tag		GTCAGTGA_TCGCTCTTACAAGGCA_ CTAGCTCATACATC_ATCCTATCTATCCAGAC_TCTCACACGTACTC
U1* removal		TGCCTTGTAAGAGCGA_TCACTGAC
*S. mansoni* capture	Biotin	TTTTTTTTTT_ATATTAACGCCCACG
*S. mansoni* target		GATTATTTGCGAGAG_CGTGGGCGTTAATAT
*S. mansoni* U1-a*		CTCTCGCAAATAATC_CTAGCTCATACATC_ ATCCTATCTATCCAGAC_TCTCACACGTACTC

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
