# Peer review of "Development of an Automated, Non-Enzymatic Nucleic Acid Amplification Test"

_micromachines, 2021, doi:10.3390/mi12101204_

Round 1

Reviewer 1 Report

In the present work, Zimmers and colleagues reported a non-enzymatic approach, named autopilot assay to detect DNA target sequences. They used synthetic sequences for a general approach and then used S. mansoni sequence for a diagnostic proof of principle. Even though the paper includes interesting points, cannot be considered for publication in the present form. Thus I recommend resubmitting the present work after addressing the following points, to match the quality of this paper to the one already demonstrated in previous works on L-DNA and DNA amplification already published by the same authors.

  • One of the main weaknesses in the present form is the lack of robust statistics. To demonstrate that the system is high throughput, the experimental statistics reported by the authors (3 repetitions) need to be increased.
  • I strongly suggest describing the gel electrophoresis results as in your previous works. Marker in fig 3 is not indicated as well in fig.S2. I strongly suggest mentioning them in the caption and clarify. In fig 3 there is a portion (?) of a gel without a label. The organization of the panel is confusing and not straightforward for the reader.
  • I would suggest running for more than one hour at 60V such concentrated agarose gel to better separate the bands. The use of an ice bath or cold room for the electrophoretic run will avoid DNA-based structure disruption due to gel heating. I would suggest revising the quantity of DNA loaded as it seems in some cases that the lanes are overloaded. Also, in some cases, bands are not well distinguishable and, in several cases, cannot allow a proper length assignation due to their shape.
  • There Is the need for experimental evidence to support speculations in Lines 282-287
  • Lines 365-368: provide the experimental evidence of the limit of detection
  • Line 191: add evidence of the mentioned "not shown data" to the supplementary information
  • The overall experimental data are promising but need to be strengthened and require more accuracy in their collection and presentation to be comparable to more routinary techniques.
  • The overall paper needs a deep revision in its form before resubmission. It appears to be too colloquial with terms like “and so on”, “washed 3x” (maybe washed three times?), “simply”, “now satisfied (line207)”, “match up very well”, “fortunately”, “dirty” biological samples just to mention some.
  • The introduction needs specific references to the state of the art and literature. As it is written is didascalic, too colloquial, and gives a personal point of view on lab practice as, in example, lines 48-61.
  • Lines 24-32: Add some lines to specify the diseases for which NA are biomarkers and related details.
  • Please add details on sensitivity and on an appropriate up-to-date comparison with the alternative tests. Consider specifying the reproducibility of the tests mentioned.
  • Increase readability and be technical
  • D-DNA and L-DNA need a more exhaustive description in the introduction.
  • Linea 74-82: reorganize the text. The aims of the work are somehow blurred and fragmented through the text. I would suggest moving referenced text before focusing on the goal of the present paper.
  • Line 87: Texas Red Is mentioned later in the text as txr, please introduce the acronyms adequately
  • Revise and rephrase Lines 100-106
  • The materials and methods section lacks details and accuracy. Brands specifics are missing for most of the instruments and materials. For example (but not only) for agarose, stain, 3d printer, fluorometer etc. They need the same accuracy of line 86. Also, the category number is an unusual way to specify materials, please revise according to common literature
  • Most remarkably, in the materials and methods section, details on the techniques are missing as well as measurements parameters.
  • Lines 127-130: the Word "follow" has been repeated too many times, rephrase.
  • Line 139: washed 3x needs to be revised with washed three times. Revise "wash 3x" in all the text
  • Lines 198-206: Need to be deeply revised and rewritten with specificity and language property
  • In the results section, figure 5 lacks panel letters and this makes the text and figure hard to follow. Add panels and refer to them properly in the text
  • Also, in the results section, I would suggest the use of appropriate language e.g. saturation rather than "crowding effect". In Lines 269-270 Is PCR or RT-?
  • Lines 346-348: maybe you meant "test the platform on well-established sequences"?
  • An explanation of what S.mansoni is, should be added to the introduction section. Also, make clearer why the authors chose this sequence rather than others. Other details are reported in the discussion section. Reorganize the text.
  • Line 355: what happens in less "fortunate" cases? This Is supposed to be a general method, that is tailored in a diagnostic setup to be specific for S.mansoni. rephrase the paragraph and clarify the issues.
  • As the "time-expensive" issue for enzymatic assays has been raised in the introduction and an alternative has been proposed, It needs to be properly discussed with appropriate comparison.
  • Maybe lines 445-449 can be somehow resumed without too much fragmentation. E.g “ZZ participated in all the phases of the work” etc.

Author Response

We would like to thank the reviewers for their thorough review and feedback for our manuscript “Development of an automated, non-enzymatic nucleic acid amplification test.” In response to the comments, we have made changes to the manuscript and now resubmit for your consideration. A complete list of reviewer comments (in italics) is below, as well as responses to each comment. Where applicable, passages of edited text are included (indented).

  1. One of the main weaknesses in the present form is the lack of robust statistics. To demonstrate that the system is high throughput, the experimental statistics reported by the authors (3 repetitions) need to be increased.

We have expanded our statistical analysis in several areas. First, the comparison of L-DNA and D-DNA amplification, both in Figures 6 and 7, was analyzed using two-way analysis of variance. The number of dumbbell incubations and the chirality of the DNA were the two factors, and the fluorescence was the dependent variable. The chirality of the DNA was found to insignificantly affect the fluorescence readings, using a 0.05 significance level. The following text was added to line 326:

The amplification of both enantiomers is virtually identical; analysis using two-way analysis of variance (ANOVA) found that the chirality of the dumbbells had no significant effect on the observed variance (α < 0.05).

The following was added to line 334:

Over the course of 15 dumbbell incubations, the D-DNA amplification (shown in Figure 7) was found to be identical according to two-way ANOVA, α < 0.05, regardless of whether or not L-DNA dumbbells have been included.

The amplification curves for varying target concentrations shown in Figure 8 were also analyzed with two-way ANOVA, this time using target copies and number of dumbbell incubations as the two factors. Here, both factors were found to be significant in their effect on fluorescence. The following was added to line 348:

Analysis using two-way ANOVA revealed that both target copy number and number of dumbbell incubations had significant effects on the resulting fluorescence measurements.

Additionally, we analyzed the signal ratio after 15 dumbbell incubations for both the development target and the S. mansoni target using a two-way t-test to determine that the signal ratios are not significantly different, based on a 0.05 significance level. The following was added to line 388:

The signal ratios after 15 cycles were calculated as 2.21 ± 0.09 for the original target and 2.18 ± 0.23 for the S. mansoni target, which are not significantly different based on a two-way t-test, α < 0.05.

Finally, we respectfully disagree with the notion that additional trials are necessary for this work. We believe that the number of repetitions was sufficient to show the trends which are discussed.

  1. Strongly suggest describing the gel electrophoresis as in your previous works. Marker in fig 3 is not indicated as well as in fig S2. I strongly suggest mentioning them in the caption and clarify. In fig 3 there is a portion of the gel without a label. The organization of the panel is confusing and not straightforward to the reader.

We have edited the gel images, as well as the figure caption, to make them clearer to the reader. Figures 3 and S2 have been edited, and the following text was added to the caption of Figure 3:

The same DNA ladder is used throughout, and shown at the right and left sides of each gel.

  1. I would suggest running for more than one hour at 60V such concentrated agarose gel to better separate the bands. The use of an ice bath or cold room for the electrophoretic run will avoid DNA-based structure disruption due to gel heating. I would suggest revising the quantity of DNA loaded as it seems in some cases that the lanes are overloaded. Also, in some cases, bands are not well distinguishable and, in several cases, cannot allow a proper length assignation due to their shape.

The expertise of the reviewer in gel electrophoresis is evident, and we thank them for the suggestions. We believe, however, that the presented gel images are sufficient to demonstrate the hybridization of the various DNA elements to one another.

  1. There is a need for experimental evidence to support speculations in lines 282-287.

The effect of steric hindrance on DNA binding is a previously-described phenomenon; we added a citation to support this idea: Mahshid et al. Journal of the American Chemical Society 137.50 (2015): 15596-15599.

  1. Lines 365-368: provide the experimental evidence of the limit of detection.

We did not evaluate the limit of detection for the S. mansoni DNA target; we tested 3*107 copies and compared the performance to 3*107 copies of the development target DNA. This is described in the manuscript.

  1. Line 191: add evidence of the mentioned “not shown data” to the supplementary section.

We have added a gel image (Figure S2) and accompanying description to the Supplementary section.

  1. The overall experimental data are promising but need to be strengthened and require more accuracy in their collection and presentation to be comparable to more routinary techniques.

We respectfully disagree with this comment.

  1. The overall paper needs a deep revision in its form before resubmission. It appears to be too colloquial with terms like “and so on”, “washed 3X”, “simply”, “now satisfied”, “match up very well”, “fortunately”, “dirty” biological samples just to mention some.

We have edited the manuscript to use less colloquial terms throughout. There are many instances, so they are not listed below. In particular, “washed 3X” was replaced with “washed three times”.

  1. The introduction needs specific references to the state of the art and literature. As it is written is didascalic, too colloquial, and gives a personal point of view on lab practice as, in example, lines 48-61.

We have edited the text in lines 49-53 of the introduction in an effort to read as less colloquial:

Due to the single-stranded complementary binding domains, combining the two dumbbells in solution will cause uncontrolled binding to one another, rather than controlled accumulation on the surface of the magnetic beads. Therefore, the dumbbells must be added to the beads one at a time, with wash steps in between each incubation.

We believe that the references in the Introduction are appropriate, and every statement requiring references has them.

  1. Lines 24-32: Add some lines to specify the diseases for which NA are biomarkers and related details.

We believe that these details are very well-established and, given the length of the manuscript, not necessary here.

  1. Please add details on sensitivity and on an appropriate up-to-date comparison with the alternative tests. Consider specifying the reproducibility of the tests mentioned.

We found this comment somewhat unclear; if the alternative test mentioned here refers to the dumbbell DNA assay reported by Xu et. al, we believe we have already provided sufficient details in the manuscript. If referring to other nucleic acid tests in general, such as PCR or LAMP, these details are well-established in the literature and we do not believe it necessary to restate them here.

  1. Increase readability and be technical.

We believe this has been covered by our edits in response to other specific comments.

  1. D-DNA and L-DNA need a more exhaustive description in the introduction.

We believe that the section on L-DNA in the introduction covers all the necessary information, as well as provides references for additional information. Given the length of the manuscript, we opted to be as succinct as possible.

  1. Lines 74-82: recognize the text. The aims of the work are somehow blurred and fragmented through the text. I would suggest moving referenced text before focusing on the goal of the present paper.

Paragraph 3 of the introduction has been edited to more clearly outline the two challenges regarding the original dumbbell DNA assay, the solutions to which are outlined in the following two paragraphs. We believe this helps to clarify the goals of the paper and the proposed solutions. The following was added to line 58:

These two obstacles, the undesirably high number of hands-on steps and the lack of an internal negative control, make the dumbbell DNA assay impractical for diagnostic applications.

  1. Line 87: Texas Red is mentioned later as TXR, please introduce the acronyms adequately.

This change has been implemented.

  1. Revise and rephrase lines 100-106.

We find the text in lines 100-106 to be clear; is there something specific that the reviewer finds unclear?

  1. The materials and methods section lacks detail and accuracy. Brands specifics are missing for most of the instruments and materials. For example (but not only) for agarose, stain, 3D printer, fluorometer, etc. They need the same accuracy as line 86. Also, the category number is an unusual way to specify materials, please revise according to common literature.

The materials and methods section has been edited to include more product info. Additional info was provided for: the autoPiLOT tactile button, the IR sensors/receivers, the 3D-printer, the agarose, the GelRed stain, the Qiagen Rotor-Gene, biotin, BSA, and Tween 20. Additionally, catalog numbers were removed from descriptions.

  1. Most remarkably, in the materials and methods section, details on the techniques are missing as well as measurement parameters.

We respectfully disagree with this comment; the provided details should be sufficient for anyone to replicate the performed experiments. It is unclear if there is a specific technique the reviewer is referring to here.

  1. Lines 127-130: the word “follow” has been used too many times. Rephrase.

The following edit has been made:

To form the double-stranded dumbbell structures, the two oligonucleotides (for example, U1-a and U1-b form dumbbell U1) were combined at equal molar ratios to the desired final concentration and then heated with the following thermal profile: 95 °C for 5 minutes, followed by 50 °C for 10 minutes, and finally by 37 °C for 10 minutes.

  1. Line 139: washed 3X needs to be revised with washed three times.

See comment 8.

  1. Lines 198-206: Need to be deeply revised and rewritten with specificity and language property.

It is unclear whether the reviewer has a specific issue with the text here. We revised the passage to read as follows:

Next, the binding of the dumbbells U1 and U2 to one another was examined. Figure 3B shows that once combined at equal ratios, the dumbbells form products of varying sizes. Distinct bands corresponding to complexes of 2, 3, and 4 dumbbells can be seen; the bands resulting from even larger complexes blend together to form a smear on the gel. This test was duplicated using FAM-labeled DNA to examine whether the presence of fluorescent labels had any apparent effect on dumbbell binding. The results (Figure 3B, right) show that the FAM-labeled dumbbells hybridized in a similar fashion as the unlabeled dumbbells.

  1. In the results section, figure 5 lacks panel letters and this makes the text and figure hard to follow. Add panels and refer to them properly in the text.

We have added letters A, B, and C to the three sections of Figure 5, and revised the caption accordingly. References to Figure 5 in the text have been updated to say 5A, 5B, or 5C.

  1. Also, in the results section, I would suggest the use of appropriate language e.g. saturation rather than “crowding effect”. In lines 269-270 is PCR or RT-?

The term “crowding effect” has been replaced with steric hindrance:

One possible explanation for this discrepancy is that when the beads already have a large amount of their surfaces covered in target DNA, the binding of additional DNA is inhibited by steric hindrance.

Also, “PCR” was replaced with “quantitative PCR” in line 292 to reflect the use of real-time fluorescence measurements.

  1. Lines 346-348: maybe you meant “test the platform on well-established sequences”?

We agree that the suggested wording is clearer; the text has been revised to read:

The previous experiments have all used the originally-published development target sequence (see Table 1, target) to test the autoPiLOT platform on well-established DNA sequences.

  1. An explanation of what S. mansoni is should be added to the introduction section. Also, make clearer why the authors chose this sequence rather than others. Other details are reported in the discussion section. Reorganize the text.

We have taken the reviewers advice and reorganized the text to introduce schistosomiasis in the introduction. The text in section 3.7 providing background on schistosomiasis and the text in the discussion providing the PCR limit of detection were removed from those sections and combined into a new final paragraph in the introduction:

Finally, as a proof-of-principle of the potential diagnostic applications for this automated, non-enzymatic DNA amplification reaction, we demonstrated the detection of Schistosoma mansoni DNA. S. mansoni is the most widespread member of the family of parasitic flatworms which cause schistosomiasis, a leading neglected tropical disease responsible for the loss of 4.5 million disability-adjusted life years (DALYs). The target sequence is part of a 121-bp tandem repeat sequence which comprises roughly 12% of the S. mansoni genome, and has previously been targeted with PCR assays. The PCR limit of detection was reported to be 1.28 pg/mL; given the highly-repeated nature of the target sequence in the genome, this corresponds to approximately 790,000 copies/mL of the target sequence.

  1. Line 355: what happens in less “fortunate” cases? This is supposed to be a general method that is tailored to a diagnostic setup to be specific to S. mansoni, rephrase the paragraph and clarify the issues.

We have clarified the language here. The dumbbell amplification assay can be repurposed to a new target sequence without changing the primary dumbbell sequences (U1 and U2). The line was edited to read:

The stepwise binding motif of the dumbbell DNA assay means that dumbbells U1 and U2 do not need to be redesigned to integrate a new target sequence; only the target-binding domain on U1-a* and the capture sequence need to be changed to bind the desired target.

  1. As the “time-expensive” issue for enzymatic assays has been raised in the introduction and an alternative has been proposed, it needs to be properly discussed with appropriate comparison.

A paragraph was added to the discussion section to discuss this point:

Reaction automation in the autoPiLOT removes the need for many hands-on pipetting steps, but does not by default decrease the total reaction time. Future research may also investigate shortening the overall autoPiLOT reaction time. We have shown (in Figure S10) that the incubation period in each dumbbell can be greatly reduced while still observing signal amplification. The exact effect on signal-to-noise ratio, however, remains unknown. Further testing of several incubation times to determine a balance between reaction time and sensitivity would be useful for diagnostic applications in which time is limited. Additionally, a more flexible design in which the reaction runs only as long as necessary could be developed; samples with higher target concentrations would amplify quicker, terminating the reaction quicker than samples with less target.

  1. maybe lines 445-449 can be somehow resumed without too much fragmentation. E.g. “ZZ participated in all the phases of the work” etc.

We have followed the guidelines laid out by the journal for this section.

Reviewer 2 Report

Dear Author, 

I enjoyed the reading of your manuscript and it's very well written. The scientific soundness is very strong.

Maybe some improvement in the graphs would be helpful. The data points size should be increased and it should be coded with different shapes (round, square, triangular etc.) so that the black and white printout of the paper can also be easily understandable. 

So far the methodology is not tested with the real urine sample so it may cause the loss of sensitivity with the real sample. Authors may discuss it and mention what is possible to do to improve the sensitivity. 

Some characterization of magnetic beads can be added so that it helps to understand the binding efficiency. 

Overall the manuscript is very informative and discusses the cost-effective methodology to be used for DNA testing. 

Thank you 

Author Response

We would like to thank the reviewers for their thorough review and feedback for our manuscript “Development of an automated, non-enzymatic nucleic acid amplification test.” In response to the comments, we have made changes to the manuscript and now resubmit for your consideration. A complete list of reviewer comments (in italics) is below, as well as responses to each comment. Where applicable, passages of edited text are included (indented).

  1. Maybe some improvements in the graphs would be helpful. The data points size should be increased and it should be coded with different shapes (round, square, triangular, etc) so that the black and white printout can also be easily understandable.

We have updated the graphs to use larger data points with different shapes to code different reaction types (squares vs diamonds, etc). Figures 5, 6, 7, and 8 were updated.

  1. So far, the methodology is not tested with the real urine sample so it may cause the loss of sensitivity with the real sample. Authors may discuss it and mention what is possible to do to improve the sensitivity.

This is a good point; we have added a brief discussion to the discussion section, lines 465-471:

It is possible that reaction sensitivity will decrease when applied to DNA extracted from urine, since urine is known to contain inhibitors such as nucleases. In the case of nucleases, heating the urine to 75 °C for 10 minutes has been shown to deactivate nucleases in urine. Sample-prep steps such as heating are likely required to maximize extraction efficiency from urine. Losses due to extraction efficiency may be countered by increasing sample volume; large volumes are easily obtained.

  1. Some characterization of magnetic beads can be added so that it helps to understand the binding efficiency.

Our group has previously characterized nucleic acid capture via magnetic beads; we have edited the manuscript to cite this paper and include a rationale of why 30 minutes was chosen as the incubation time. The following can be found in lines 211-214:

Sequence-specific nucleic acid capture via functionalized magnetic beads has previously been shown to be a slow process, with low efficiency for incubation times shorter than 30 minutes. Therefore, 30 minutes was chosen as the incubation time.

Reviewer 3 Report

The paper describes a test for non-enzymatic nucleic acid amplification. This paper need to be improved in order to clarify the methods, and especially for clarifying the novelty. I have the following comments, suggestions and questions.

1) The authors claim an automated method. However, there are many manual or independent steps during the whole process. For example, Is the dumbbell formation a part of the automatic process? Are the temperatures 95ºC, 50ºC and 30ºC controlled with the automatic device?. It is not clear. I would suggest to clarify the automatic steps and the manual steps, or at least, clarify the first step of the automatic process and the final one.

2) The proposed device seems to be very similar to the ones reported on references 22 and 23. Could the authors comments the improvement of the proposed device over the previously reported [ref 22 and 23]. If this is one of the most important results of the paper, I would suggest to include it in the manuscript (removing from the supplementary materials).

3) The device is commented in section 2.2 and section 3.3. Is it possible to merge these sections?.

4) Regarding to section 2.2:

  • Could the authors include the model of the motors?, and define the QRD1114 as a reflective object sensor?
  • The analog input A0 is avaliable in the "motor arduino". What is the reason for using two arduinos?
  • The value of the resistor (217 ohm) is not standard. Is it necessary this value? Why do you use that value?

5) Are the "air valves" of the 'figure 2' "air gaps"? If so, please modify the figure 2 including "air gaps".

6) How long does the whole amplification test take?. What is the velocity of the liquids during the movement?

7) Regarding the "discussion". It is not necessary to comment the colorimetric process in detail.

8) Please, include a conclusion section, and use the "discussion" to comment the experimental results. The manuscript is very long and a conclusion section is important.

Minor comments and questions:

1) Please extend the acronyms TE, TBE, TXR and autoPilot. The last one in the abstract.

2) The first reference is Ref 2. Please check

3) Is the diameter of the tube 3/32"? It is very far from microfluidics.

4) Figure 1 is commented twice, in the introduction and in section 3.1

5) Could you include "Gel1" and "Gel2" in the figure 3A?

6) Are the lines 2 and 5 of Gel1 of Figure3A for U1-b?. (Just to confirm)

7) Curiosity: Why did the authors use a high percentage agarose gel 3%?

Author Response

We would like to thank the reviewers for their thorough review and feedback for our manuscript “Development of an automated, non-enzymatic nucleic acid amplification test.” In response to the comments, we have made changes to the manuscript and now resubmit for your consideration. A complete list of reviewer comments (in italics) is below, as well as responses to each comment. Where applicable, passages of edited text are included (indented).

  1. The authors claim an automated method. However, there are many manual or independent steps during the whole process. For example, is the dumbbell formation a part of the automatic process? Are the temperatures 90, 50, and 30 C controlled with the automatic device? It is not clear. I would suggest to clarify the automatic steps and the manual steps, or at least, clarify the first step of the automatic process and the final one.

We have clarified this with the following insertion into line 166:

All reaction steps after the incubation with dumbbell U1* are therefore automated.

Line 218 was also edited to clarify:

Incubations in dumbbells U1 and U2 were then performed automatically using the autoPiLOT reaction processor.

  1. The proposed device seems to be very similar to the one reported in references 22 and 23. Could the authors comment the improvement on the proposed device over the previously reported (ref 22 and 23). If this is one of the most important results of the paper, I would suggest to include it in the manuscript (removing from supplementary materials).

The primary improvement over previous devices is the automated control from the Raspberry Pi. Additionally, previous devices have only ever demonstrated one-directional movement of magnetic beads in a linear fashion. Finally, previous devices have not detected liquid/air interfaces like the autoPiLOT, necessitating precise loading of the tubing. We have attempted to explain these innovations in the subsequent paragraphs, and have edited the following to line 137 to clarify the limitations of earlier devices:

Rotation of the gears therefore moves the microfluidic tubing which contains the various reaction fluids, as previously described for one-directional sample prep devices.

  1. The device is commented in section 2.2. and section 3.3. Is it possible to merge these sections?

Yes, we have eliminated former section 3.3 and merged that information into section 2.2.

  1. Regarding section 2.2:
  • Could the authors include the model of the motors? And define the QRD1114 as a reflective object sensor?

Yes, we have made these changes.

  • The analog input A0 is available in the “motor arduino”. What is the reason for using two arduinos?

A single Arduino Uno was not able to accurately report the QRD1114 sensor value at a high sampling rate while monitoring and parsing through serial inputs from the Raspberry Pi. Consolidating both functions to one Arduino was attempted originally but issues with analog input voltage drift, missed motor commands, and dropped serial connections led to the two-Arduino design. Future revisions could optimize communications and memory management to reduce the hardware needs to one microcontroller.

  • The value of the resistor (217 ohm) is not standard. Is it necessary this value? Why do you use that value?

217ohms is not a necessary value. The current-limiting resistor for the QRD1114 emitter LED can be set from 175ohms to 330ohms depending on the desired output current. This device was built remotely during the COVID-19 pandemic and as such available resistors were combined in parallel to achieve the lowest resistance within that range: 217ohms. The air gap-sensing algorithm was calibrated using input values from the QRD1114 with a 217ohm current-limiting resistor, so we did not change this value later. A more standard resistor choice would only need subtle adjustment to the air gap sensing code.

  1. Are the “air valves” of figure 2 “air gaps”? If so, please modify the figure 2 including “air gaps”.

They are indeed air gaps; the figure has been modified.

  1. How long does the whole amplification test take? What is the velocity of the liquids during the movement?

We have revised the materials and methods to include this information. The following excerpts are now in lines 138 and 229, respectively:

The reaction cassette was moved at a speed of 0.2 cm/s to transport the magnetic beads, and 7.5 cm/s to break the beads out of the magnetic field.

The total reaction time was a function of the number of dumbbell incubations; 15 incubations on the autoPiLOT took approximately 8 hours (15 incubations x 0.5 hours/incubation, plus the time to move the beads back and forth between chambers).

  1. Regarding the “discussion”. It is not necessary to comment the colorimetric process in detail.

We have simplified the description of the colorimetric readout.

  1. Please include a conclusion section, and use the discussion to comment the experimental results. The manuscript is very long and the conclusion section is important.

We have added a brief conclusion paragraph.

The autoPiLOT platform has automated a previously reported dumbbell DNA amplification scheme. In addition to automation, the addition of parallel L-DNA dumbbell amplification components demonstrated the utility of left-handed L-DNA to enable simultaneous measurement both specific and non-specific signals in the same reaction.

Minor comments/questions:

  1. please extend the acronyms TE, TBE, TXR, and autoPiLOT. The last one in abstract.

We have edited accordingly.

  1. the first reference is ref 2. Please check.

Apparently, the inclusion of a reference in the caption of Figure 1 produced this error. This has been corrected.

  1. Is the diameter of the tube 3/32”? It is very far from microfluidics.

Yes, the inner-diameter is 3/32” (approximately 2.3 mm). We refer to this as microfluidic because the diameter is small enough that microfluidic phenomena, such as the surface tension of the fluids holding the liquid/air interfaces intact, have begun to take over.

  1. Figure 1 is commented twice, in the introduction and in section 3.1.

Yes, we believe it is helpful in section 3.1 to reference Figure 1 again.

  1. could you include “gel 1” and “gel 2” in Figure 3A?

Yes, the figure is updated.

  1. Are the lines 2 and 5 of gel 1 of figure 3A for U1-b? (just to confirm)

Yes, it was displayed twice since it is a component of both dumbbells U1* and U1.

  1. Curiosity: why did the authors use a high percentage agarose gel 3%?

The 3% gel causes the DNA to move more slowly, and produced better results for such short DNA fragments.

Round 2

Reviewer 1 Report

The authors have partially answered the questions and clarified the concepts. The paper has been improved from the first submission. However, the reviewer believes that the manuscript can still be improved. As it is presented, the conclusion section is too short, and part of the discussion is not commenting on the results therefore I strongly suggest rearranging the paragraphs. Also, for the details that are very well-established, and considered not necessary by the author I still recommend adding the appropriate reference to the text. Also, some experiments such as additional gel electrophoresis have not been provided (for example it would be appreciated to show to the reviewer at least some titration experiments). However, the manuscript, with some additional efforts, can reach the quality of the previous paper published by the authors.

Author Response

Thank you for the continued feedback on our manuscript “Development of an automated, non-enzymatic nucleic acid amplification test.” We have made further edits to the manuscript based on your feedback, primarily to the Conclusion section. Some material covered in the Discussion section has been moved into the Conclusion, so that the Discussion is focused on interpretation of results and direction of future research to follow up on un-answered questions or significant improvements. The Conclusion section is now longer and provides a more thorough summary of the most significant findings and the impact of the work presented here. The new Conclusion is pasted below:

The dumbbell DNA amplification scheme has previously been shown to be a highly-sensitive, non-enzymatic method of detecting target DNA. The largest obstacles to its use in diagnostic applications were the extremely high number of manual pipetting steps and hands-on time required, and the need for an additional control reaction. The autoPiLOT platform has overcome these obstacles through automation of the reaction in a self-contained, microfluidic reaction cassette. The components are primarily 3D-printed parts and low-cost electronics, making the autoPiLOT a low-resource compatible diagnostic platform.

The autoPiLOT assay also demonstrates the utility of left-handed L-DNA as an internal control. Whereas a traditional assay format would require the performance of two parallel reactions, one with and one without target, for interpretation of results, the dual- chirality design used here discretely measured both specific and non-specific signals in the same reaction. In addition to saving time and reagents, this opens the door to testing unpurified biological samples which may exhibit varying rates of non-specific amplification that cannot be accurately simulated with a parallel control reaction.

Reviewer 3 Report

The authors have answered the questions, and clarified the required concepts. The paper has been improved. However, I would like the authors to improve the conclusion section and the abstract following these comments.

1) The Conclusion section is too short. It is not valid for a journal conclusion section. In addition, the Discussion is a combination of Discussion and Conclusions.

The Discussion is used to comment (discuss) the experimental results, and the Conclusion is used as a brief summary and to comment general issues of the paper (for example advantages and limitations of the proposed device) and future work.

For example, in my opinion, among others, the following paragraphs of the "Discussion" are not commenting the experimental results:

A. "The dumbbell DNA amplification scheme has previously been shown to be a highly sensitive, non-enzymatic method of detecting target DNA [15]. The largest obstacles to its use in diagnostic applications were the extremely high number of manual pipetting steps and hands-on time required, and the need for an additional control reaction. The autoPiLOT platform has overcome these obstacles through automation of the reaction in a self-contained, microfluidic reaction cassette. The components are primarily 3D-printed parts and  low-cost  electronics,  making  the  autoPiLOT  a  low-resource  compatible  diagnostic platform. Real-time fluorescence measurements were shown to reflect the amount of DNA which had accumulated on the magnetic beads"

B. "Future work should focus on optimization of the autoPiLOT magnetic bead control strategies, in an effort to more efficiently bind dumbbells and achieve exponential amplification."

C. "Future  research  should  also  work  toward  the  implementation of a  lower-cost fluorescence imaging system.  The fluorometer  used in the  autoPiLOT  is the  only expensive component, with a price tag of several thousand dollars. Recently, smartphone-based fluorescence microscopes have grown in popularity and sophistication, bringing high quality digital cameras and multifunctionality to the point-of-care [31,32]. It is estimated that approximately 80% of the world population uses smartphones, making a smartphone optical readout highly applicable, even in low-resource settings [33]".

D. "Reaction automation in the autoPiLOT removes the need for many hands-on pipetting steps, but does not by default decrease the total reaction time.  Future research may also investigate shortening the overall autoPiLOT reaction time."

Could the authors organise and re-write the Discussion and Conclusion sections following this suggestion?

2) The total reaction time is an interesting parameter of any process. Please, include it in the Abtract

Author Response

(The authors gave the same response as above.)
